# VisionFocus: Towards Efficient Hallucination Mitigation via Token-Aware Visual Enhancement

## Abstract

Despite their impressive capabilities, Multimodal Large Language Models (MLLMs) are prone to hallucinations. Recent efforts to address this issue have primarily focused on suppressing the inherent language priors of Large Language Models (LLMs) through contrastive decoding or uniformly enhancing attention to all visual tokens via attention intervention. However, these approaches either incur significant inference latency or exacerbate hallucinations in certain cases. In this work, we identify a critical insight: *Not all visual tokens are beneficial for hallucination mitigation*. Specifically, we observe that the vision encoder in MLLMs gradually focuses its attention on a limited subset of visual tokens. Further experiments demonstrate that tokens receiving high attention are crucial for mitigating hallucinations, whereas indiscriminate enhancement of low-attention tokens may exacerbate them. Based on these findings, we propose **VisionFocus**, a training-free, efficient, and plug-and-play method to mitigate hallucinations. It guides the model to concentrate on informative visual tokens during decoding, while avoiding excessive amplification of irrelevant or distracting visual information. This selective enhancement strengthens visual grounding and effectively mitigates hallucinations. Extensive experiments on six widely used benchmarks demonstrate the effectiveness of VisionFocus in mitigating hallucinations across various MLLM families without requiring additional training. In addition, VisionFocus achieves state-of-the-art performance in hallucination mitigation while maintaining competitive decoding speed, highlighting its practical utility. The code and models will be made publicly available soon.

## 1 Introduction

With the development of Large Language Models (LLMs) (Bai et al., 2023; Zhu et al., 2023b), Multimodal Large Language Models (MLLMs) (Bai et al., 2025; Liu et al., 2024a;b) have rapidly emerged. MLLMs, known for their ability to process multimodal data such as images, audio, and text, have been widely applied in various fields, including natural language processing (Tu et al., 2023) and computer vision (Koh et al., 2023).

However, the presence of hallucination poses a significant challenge to the practical application of MLLMs (Huang et al., 2024; Bai et al., 2024), as they might generate output that diverges from the inputs, including generating imaginary objects or offering conflicting assessments. This issue notably undermines the reliability of MLLMs, especially in safety-sensitive domains like healthcare (Lin et al., 2025) and autonomous driving (Ding et al., 2024).

Recent research efforts (Leng et al., 2024; An et al., 2025; Wan et al., 2025) have aimed to mitigate hallucinations by employing Contrastive Decoding (CD) to suppress the built-in language biases of LLMs. While CD approaches show promise due to their training-free paradigm and competitive performance, they introduce notable inference delays because of the need for multi-round inference (Fig. 1(a)). This could impede their scalability and practical utility, especially in environments with limited computational resources.

On the other hand, attention intervention techniques (Yin et al., 2025; Liu et al., 2024c) circumvent significant inference delays but face different challenges. These approaches attribute hallucinations

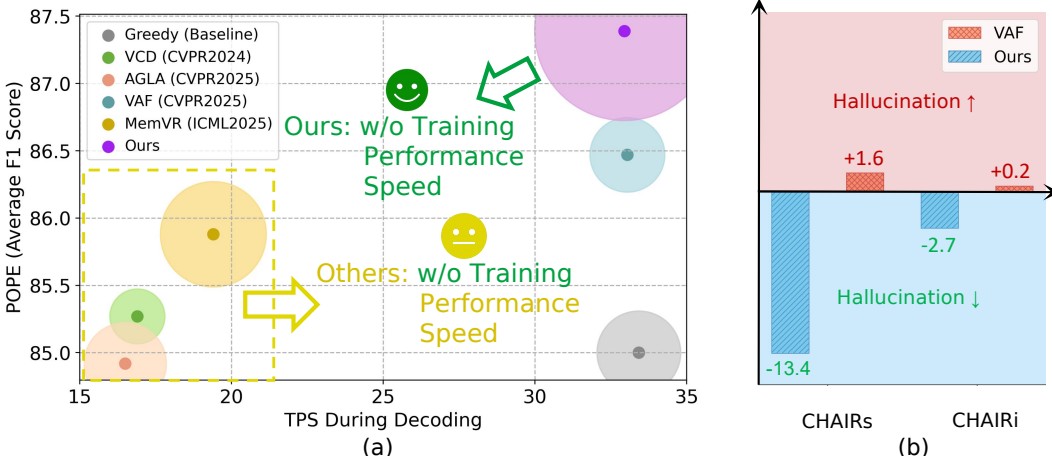

Figure 1: **Performance comparison of various methods on LLaVA-1.5.** **(a)** A higher average F1-score on POPE indicates fewer hallucinations, while a higher Tokens Per Second (TPS) reflects better decoding efficiency. Larger bubble sizes indicate lower hallucination rates on CHAIR. Comparatively, our method achieves both higher efficiency and lower hallucinations. **(b)** Performance of VisionFocus (Ours) and attention intervention methods (e.g., VAF (Yin et al., 2025)) on CHAIR, compared to the vanilla LLaVA-1.5. Results indicate that attention intervention methods may exacerbate hallucinations in certain scenarios.

to inadequate attention given to visual tokens during decoding and typically address this by uniformly enhancing the attention weights of all visual tokens. However, as illustrated in Fig. 1(b), such uniform enhancement could potentially worsen hallucination issues. This prompts a crucial question: *Are all visual tokens indispensable for alleviating hallucinations?*

**Key Observations.** To examine the impact of various visual tokens on mitigating hallucinations, we first scrutinize the attention distribution in the visual encoder of MLLMs. The analysis in Fig. 2(a) shows a distinct focus of attention on a restricted subset of visual tokens. Subsequent experiments (Fig. 2(b)) show that high-attention visual tokens play a vital role in reducing hallucinations, as they typically aggregate global visual information (details in Sec. 2.3). In contrast, indiscriminate enhancement of low-attention visual tokens may worsen hallucinations. Therefore, *a strategy for selective visual enhancement is imperative for mitigating hallucinations more effectively.*

**Our Solution.** Inspired by these findings, we propose VisionFocus, a training-free, efficient, and plug-and-play approach for hallucination mitigation. VisionFocus directs the model to prioritize essential visual tokens during decoding, which prevents the over-amplification of extraneous or distracting visual details. Specifically, VisionFocus consists of two modules: Selective Visual Enhancement (SVE) and Semantic Covariant Attention (SCA). SVE selectively enhances the attention scores of visual tokens that aggregate global visual information during the LLM inference, which improves both the completeness and diversity of the generated output. Moreover, based on the semantic invariance observed in the Q-K attention patterns of LLM (Fig. 4), SCA enhances the model's capability to capture precise local semantic features by using semantically rich key states as queries, thus enhancing the accuracy of fine-grained detail generation. The collaborative functioning of SVE and SCA ensures the model's output is both comprehensive and accurate.

As shown in Fig. 1(a), VisionFocus achieves the best performance in hallucination mitigation while maintaining high decoding efficiency. Our method incurs minimal (only 1.3%) extra latency overhead over the baseline, significantly lower than typical CD-based approaches such as VCD (Leng et al., 2024) and AGLA (An et al., 2025), which approximately double the baseline's decoding latency (see Appendix C.3 for details). Comprehensive experiments on hallucination-specific and general-purpose benchmarks demonstrate that VisionFocus consistently improves accuracy while incurring minimal computational overhead.

To summarize, our contributions are as follows:

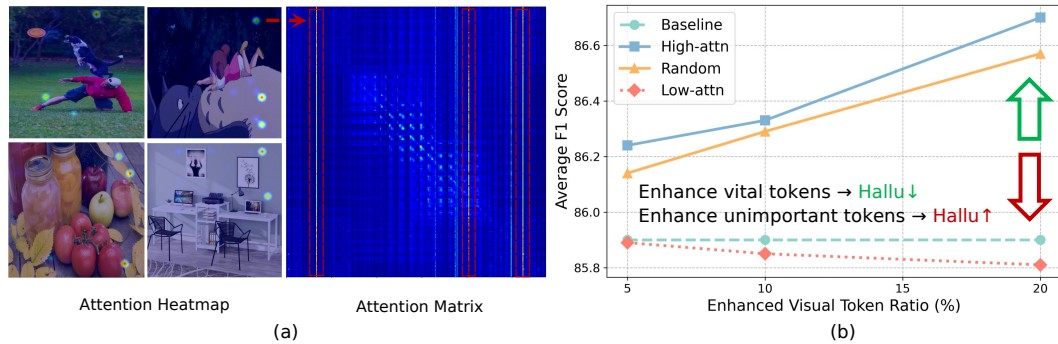

Figure 2: **Motivation of VisionFocus. (a)** Left: Attention distribution of the [CLS] token to visual tokens. Right: Attention distribution between visual tokens. **(b)** Effect of enhancing visual tokens with varying attention scores on POPE. Results show that high-attention tokens are vital for mitigating hallucinations, while indiscriminately enhancing low-attention tokens may exacerbate them.

- We investigate the attention distribution in the visual encoder of MLLMs and identify a distinct focus of attention. Our experiments suggest that not all visual tokens contribute positively to hallucination mitigation, and some may instead have adverse effects.
- We propose VisionFocus, a training-free, efficient, and plug-and-play approach that guides the model to focus on informative visual tokens, while avoiding the excessive enhancement of irrelevant or distracting visual information that could induce hallucinations.
- Comprehensive experiments demonstrate that VisionFocus outperforms existing state-of-the-art methods in hallucination mitigation. Results on general-purpose benchmarks and multiple MLLMs highlight the strong generalization ability of our method.

## 2 PRELIMINARIES

In this section, we briefly introduce the basic concepts and works relevant to this study in Sec. 2.1 and Sec. 2.2, respectively, establishing the necessary background. Following this, in Sec. 2.3, we outline our key observations and the insights behind our proposed method.

### 2.1 MLLMS AND CHALLENGES

In recent years, MLLMs have advanced rapidly, building on the foundations of earlier Vision-Language Models (VLMs). CLIP (Radford et al., 2021), as a representative early VLM, uses separate visual and textual encoders to extract unimodal features, integrating the modalities via a simple inner product. This approach enables basic cross-modal understanding but is limited in its reasoning capabilities. The advent of open-source LLMs (Bai et al., 2023; Zhu et al., 2023b), such as the LLaMA series (Touvron et al., 2023), has catalyzed a paradigm shift by introducing LLMs as the core modality fusion component. This integration has significantly enhanced multimodal reasoning and comprehension. Recent MLLMs, such as LLaVA-NeXT (Liu et al., 2024b) and Qwen2.5-VL (Bai et al., 2025), further improve cross-modal alignment through a modular architecture that connects visual encoders to language backbones via feature projection layers or intermediate modules such as Q-Former (Huang et al., 2023).

However, hallucinations remain a problem in MLLMs despite these advancements. Resolving them is crucial for ensuring the reliability of MLLMs in practical applications.

### 2.2 MITIGATING HALLUCINATIONS IN MLLMS

Researchers have made extensive efforts to reduce hallucinations. Existing methods are typically categorized into two main paradigms: training-based and training-free approaches. Training-based pipelines, such as Supervised Fine-Tuning (SFT) (Yu et al., 2024; Tang et al., 2024; Zhang et al., 2024), Reinforcement Learning (RL) (Ben-Kish et al., 2023; Kim et al., 2024; Jing & Du, 2024),

and Direct Preference Optimization (DPO) (Xiao et al., 2025; Ouali et al., 2024; Zhou et al., 2024), require access to high-quality labeled datasets and substantial computational resources.

In contrast, training-free approaches require no additional model training and primarily focus on strategies such as hidden states intervention (Wang et al., 2025; Jiang et al., 2024), CD (Wang et al., 2024b; Wan et al., 2025; Huo et al., 2024), and attention intervention (Liu et al., 2024c; Tu et al., 2025; Yin et al., 2025; Arif et al., 2025).

**Discussion.** Though training-based approaches are effective, their scalability is constrained by the large amounts of high-quality data and significant computational resources they demand. In contrast, CD and attention intervention, as mainstream training-free approaches, require fewer resources while maintaining competitive performance. However, the two-stage inference in CD leads to notable inference latency. On the other hand, attention intervention avoids the inference delay of two-stage decoding but uniformly boosts attention across all visual tokens, potentially worsening hallucinations stemming from the less informative ones. Although PAINT (Arif et al., 2025) attempts to only enhance vital visual tokens, it still assigns identical enhancement weights to all selected tokens, which neglects the differences between individual visual tokens.

## 2.3 KEY OBSERVATIONS

To examine the roles of different visual tokens during model inference, we conduct a preliminary analysis of the visual tokens produced by the vision encoder of LLaVA-1.5 (Liu et al., 2024a).

Specifically, we randomly sample some images and visualize each visual token's attention received from the [CLS] token (Fig. 2(a), Left) in the penultimate layer, which is commonly used to extract visual tokens in most MLLMs. Since the [CLS] token is typically trained for image or image-text classification tasks (Dosovitskiy et al., 2020; Park & Kim, 2022), the visual tokens it attends to can capture global semantics, playing a crucial role in image comprehension. In addition, we also visualize each token's attention received from the remaining visual tokens in the same layer (Fig. 2(a), Right), which reveals the information flow among visual tokens.

As shown in Fig. 2(a), the attention from the [CLS] token to visual tokens, and the attention among visual tokens themselves, are both concentrated on a limited number of tokens. The majority of visual tokens receive minimal attention, and only a few tokens exhibit comparatively higher attention weights. Additional visualizations of different layers and vision encoders are presented in Appendix A.1. This observation indicates that *the visual tokens with high attention scores may play a different role in hallucination mitigation compared to those with low attention scores.*

**Not All Visual Tokens Matter.** To explore the relationship between the attention levels of visual tokens and their impact on mitigating hallucinations, we conduct an analysis using the POPE benchmark (Li et al., 2023). Concretely, we arrange the visual tokens in descending order according to the attention scores assigned by the [CLS] token in the penultimate layer of the vision encoder. Subsequently, we choose a consistent portion of tokens using three methods: random selection, top-ranking (High-attn), and bottom-ranking (Low-attn). We then employ attention enhancement on the chosen visual tokens following that of VAF (Yin et al., 2025), and compare the result with vanilla LLaVA-1.5 (Baseline). Further implementation details are provided in Appendix A.2.

Results in Fig. 2(b) demonstrate that enhancing attention to high-attention visual tokens leads to the most effective hallucination mitigation, outperforming both the random strategy and the baseline. This is because tokens receiving significant attention typically aggregate global visual information (Yang et al., 2025). In contrast, enhancing low-attention tokens tends to exacerbate hallucinations. The results imply that not all visual tokens contribute positively to the model's reasoning, and some may even introduce negative effects. This highlights the need for a selective visual enhancement strategy to effectively mitigate hallucinations.

## 3 METHOD

### 3.1 OVERVIEW

To address the issues discussed in Sec. 2, we introduce VisionFocus, a training-free and efficient method designed to guide the model to focus on informative tokens during inference, while avoiding

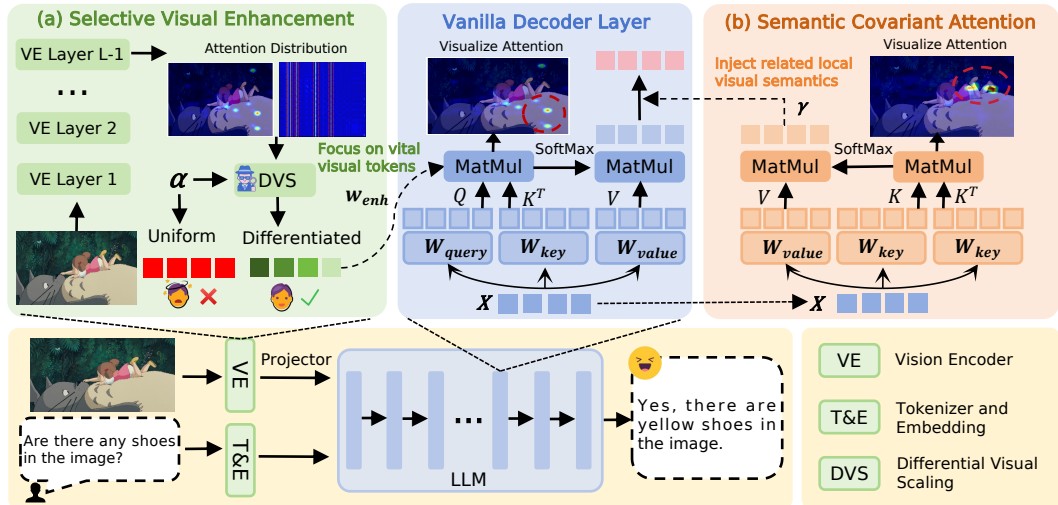

Figure 3: **Architecture of VisionFocus**. VisionFocus consists of two modules: **(a)** Selective Visual Enhancement (SVE) and **(b)** Semantic Covariant Attention (SCA). SVE selectively enhances the attention score of vital visual tokens that aggregate global visual information, while SCA strengthens the model's ability to capture fine-grained local semantic features by using key states as queries.

excessive enhancement of distracting ones. As illustrated in Fig. 3, VisionFocus consists of two modules: *Selective Visual Enhancement (SVE)* and *Semantic Covariant Attention (SCA)*.

Specifically, during LLM inference, SVE (Fig. 3(a)) selectively enhances the attention scores of critical visual tokens that aggregate global visual information, guided by the attention distribution in the vision encoder. This module improves both the completeness and diversity of the generated output. Differently, SCA (Fig. 3(b)) strengthens the model's ability to capture fine-grained local semantic features by utilizing key states that contain semantic information as queries. This enables the model to generate fine-grained details more accurately. Their synergistic integration enables the model to generate outputs that are not only comprehensive but also highly precise.

### 3.2 SELECTIVE VISUAL ENHANCEMENT

**Motivation.** In Sec. 2.2, we reveal that existing attention intervention methods may exacerbate hallucinations. In Sec. 2.3, we further observe that not all visual tokens contribute positively to hallucination mitigation, and some may amplify hallucinations. Motivated by these findings, we propose SVE, which selectively enhances attention to globally representative visual tokens while keeping uninformative ones unchanged or suppressing them. SVE consists of two components: Differential Visual Scaling (DVS) and Attention Sink Calibration (ASC), as detailed below.

**Differential Visual Scaling.** Guided by the attention distribution in the vision encoder, DVS encourages the model to focus on informative visual tokens by assigning higher enhancement weights. The weights are subsequently applied to the attention computation within the LLM.

For vision encoders with a [CLS] token (e.g., CLIP), in the penultimate layer, we use the attention scores of the [CLS] token to all visual tokens to identify informative visual tokens. Let $\mathbf{Q}_{\text{CLS}} \in \mathbb{R}^{H \times 1 \times D}$ represent the query state of the [CLS] token, where $H$ denotes the number of attention heads in the vision encoder, and $D$ represents the dimension of hidden states. $\mathbf{K}_v \in \mathbb{R}^{H \times N \times D}$ corresponds to the key states of all visual tokens, where $N$ is the total number of visual tokens. We calculate $\mathbf{A}_{\text{CLS}} \in \mathbb{R}^{H \times 1 \times N}$ as follows:

$$\mathbf{A}_{\text{CLS}} = \text{SoftMax}\left(\mathbf{Q}_{\text{CLS}}\left(\mathbf{K}_v\right)^{\mathsf{T}} / \sqrt{d}\right), \tag{1}$$

where $d$ represents the dimension of each attention head. We define the average of $\mathbf{A}_{\text{CLS}}$ across the head dimension as $\hat{\mathbf{A}}_{\text{CLS}}$. On the other hand, for vision encoders without a [CLS] token (e.g., SigLIP (Zhai et al., 2023)), in the same layer, we compute the average attention each token receives

from remaining visual tokens in the sequence, denoted as $\mathbf{A}_{\text{avg}}$. The detailed implementation of $\mathbf{A}_{\text{avg}}$ is provided in Appendix B.2.

To enhance the model's attention to informative visual tokens, we define $F(\mathbf{A}, \alpha)$ as a function that assigns differentiated enhancement weights to each visual token, where $\mathbf{A}$ denotes either $\hat{\mathbf{A}}_{\text{CLS}}$ or $\mathbf{A}_{\text{avg}}$. The weights are constrained within a predefined range determined by the hyperparameter $\alpha$ ($\alpha > 0$). We define the enhancement weights $\boldsymbol{w}_{\text{enh}} \in \mathbb{R}^N$ as follows:

$$\boldsymbol{w}_{\text{enh}} = F(\mathbf{A}, \alpha). \tag{2}$$

The computed weights are subsequently applied to the attention score calculation within the LLM, amplifying the model's focus on informative visual tokens while avoiding the excessive enhancement of distracting tokens. The details of $F(\mathbf{A}, \alpha)$ are provided in Appendix B.1.

**Attention Sink Calibration.** Although the DVS enhances the model's attention to vital visual tokens, previous studies (Zhuang et al., 2025; Yin et al., 2025) have demonstrated that, during the LLM inference, excessive attention often concentrates on low-semantic system prompt tokens, a phenomenon known as attention sink. The attention sink dilutes the effectiveness of visual enhancement. To mitigate this issue, a penalty is applied to system prompt tokens to counterbalance the excessive attention, thereby optimizing the overall attention distribution and enhancing the model's perception of vital visual information.

Let $\mathbf{Q}$ and $\mathbf{K}$ denote the query and key embeddings of the LLM, respectively. SVE modifies the attention score matrix of the LLM as follows:

$$\mathbf{Z} = \mathbf{Q}\mathbf{K}^{\mathsf{T}}/\sqrt{d}, \tag{3}$$

$$\hat{\mathbf{Z}} = \mathbf{Z} + \boldsymbol{w}_{\text{enh}} \circ \mathbf{M}_{\text{enh}} \circ \mathbf{Z} - \beta \cdot \mathbf{M}_{\text{sup}} \circ \mathbf{Z}. \tag{4}$$

Here, $\mathbf{Z} \in \mathbb{R}^{H \times L \times L}$ denotes the attention score matrix before the SoftMax operation, where $L$ represents the length of the input sequence fed into the LLM, and $H$ denotes the number of attention heads in the LLM. $\hat{\mathbf{Z}} \in \mathbb{R}^{H \times L \times L}$ denotes the attention matrix modified by SVE. The parameter $\beta$ ($0 < \beta < 1$) controls the degree of attention suppression applied to system prompts. The enhancement and suppression mask matrices, denoted as $\mathbf{M}_{\text{enh}}$ and $\mathbf{M}_{\text{sup}}$ respectively, are introduced to guide the modulation of attention weights:

$$\mathbf{M}_{\text{enh}}(i, j) = \begin{cases} 1 & \text{if } i \in \mathcal{I} \text{ and } j \in \mathcal{V} \\ 0 & \text{otherwise} \end{cases}, \tag{5}$$

$$\mathbf{M}_{\text{sup}}(i, j) = \begin{cases} 1 & \text{if } i \in \mathcal{I} \text{ and } j \in \mathcal{S} \\ 0 & \text{otherwise} \end{cases}, \tag{6}$$

where $\mathcal{V}$, $\mathcal{I}$, and $\mathcal{S}$ represent the index sets of image tokens, instruction tokens, and system prompt tokens, respectively. These modifications enhance the model's focus on informative visual tokens while reducing redundant attention to uninformative ones, ultimately reducing hallucinations.

## 3.3 SEMANTIC COVARIANT ATTENTION

**Motivation.** While SVE module helps mitigate hallucinations by guiding the MLLM to pay more attention to vital visual tokens, we still observe that the model tends to hallucinate when responding to questions that require understanding fine-grained image details. To investigate the underlying cause, we visualize the cross-modal attention from words to visual tokens during generation.

As shown in Fig. 4(b), the attention distribution between words and visual tokens in the LLM exhibits a clear pattern of *semantic invariance*. The attention, irrespective of whether the word is "shoes" or "hair", fails to capture semantically relevant visual regions and instead stays localized in specific areas of the image. These consistently attended regions correspond to the visual tokens that aggregate global visual information in the vision encoder.

This phenomenon may arise from the inherent properties of Q-K attention. Prior studies (Wang et al., 2023b; Dong et al., 2025) have shown that the LLM tends to establish the overall content framework during the generation of the first token. Consequently, in subsequent decoding steps, the query state at each step often focuses on globally representative visual tokens to maintain semantic completeness

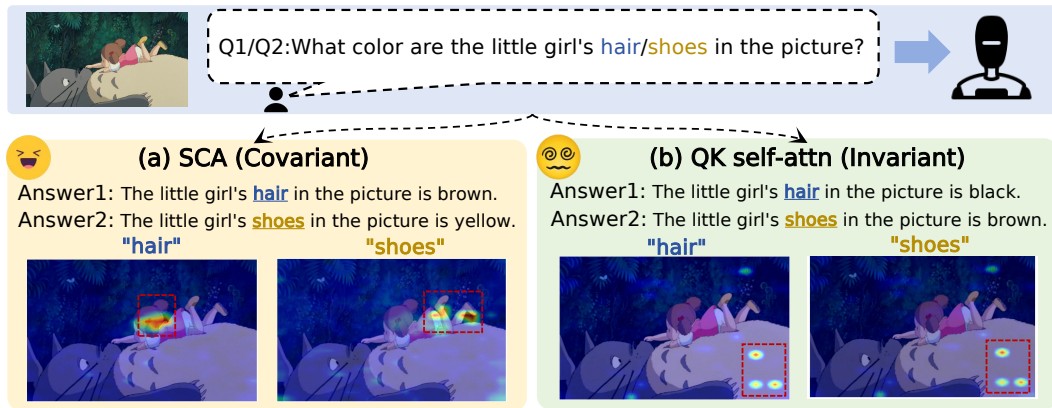

Figure 4: **Comparison of SCA and Q-K attention in localizing critical visual cues.** We visualize the attention weights from words to visual tokens for two attention mechanisms during next token generation. The results show that vanilla Q-K attention exhibits *semantic invariance*, while the attention distributions under SCA are *semantically covariant*.

and coherence. Therefore, *the Q-K attention mechanism requires modification to better localize relevant visual semantic features.*

**Local Semantic Alignment.** Based on the above observations, we hypothesize that the key state in the LLM contains the full semantic information of the associated token (Yang et al., 2025). Therefore, we propose using the key state as the query in the LLM's attention computation, enabling the model to focus more precisely on visual regions semantically correlated with the token itself. We define this mechanism as SCA, formulated as:

$$\text{Attn}_{\text{SCA}} = \text{SoftMax}\left(\mathbf{K}\mathbf{K}^{\mathsf{T}}/\sqrt{d}\right). \tag{7}$$

As shown in Fig. 4(a), replacing the query with the key state produces a significantly more accurate visual attention distribution. The model can more precisely attend to local visual regions associated with keywords such as "shoes" and "hair". Additional visualizations demonstrating the superiority of SCA over standard Q-K attention are provided in Appendix A.3. Similar behaviors have also been observed in the semantic segmentation tasks of VLMs (Wang et al., 2024a). Overall, SCA enables the LLM to dynamically adjust its attention distribution in accordance with the semantic content being generated, thereby improving the model's ability to focus on relevant local visual regions. Additional analyses of SCA's effectiveness are provided in Appendix B.3.

However, using SCA alone to guide attention toward local semantic information compromises the completeness of the generated content, thereby affecting overall output quality. To overcome this limitation, we integrate SVE and SCA, as defined below:

$$\mathbf{Q} = \text{Proj}_q(\mathbf{X}), \quad \mathbf{K} = \text{Proj}_k(\mathbf{X}), \quad \mathbf{V} = \text{Proj}_v(\mathbf{X}), \tag{8}$$

$$\mathbf{Y} = \mathbf{X} + \text{Proj}(\text{SoftMax}(\hat{\mathbf{Z}}) \cdot \mathbf{V} + \gamma \cdot \text{Attn}_{\text{SCA}} \cdot \mathbf{V}), \tag{9}$$

$$\mathbf{O} = \mathbf{Y} + \text{FFN}(\mathbf{Y}). \tag{10}$$

Here, $\mathbf{X}$ denotes the input to the LLM, while $\mathbf{Q}$, $\mathbf{K}$, and $\mathbf{V}$ represent the query, key, and value embeddings, respectively. Proj refers to the projection layers, and FFN denotes a feed-forward network. For simplicity, normalization operations are omitted. The hyperparameter $\gamma$ ($\gamma > 0$) is introduced to regulate the model's attention to relevant local visual details, thereby achieving a balanced and synergistic integration of SVE and SCA. Further details of the fusion strategy are provided in Appendix B.3.

Table 1: Performance of various methods on POPE, with the best results highlighted in **bold**. We report Accuracy (Acc) and F1-score (F1) under three settings. The symbols ↑ and ↓ denote that higher and lower values are preferred, respectively. * denotes that the authors have not released an implementation of this model, and the reported results are obtained from our reimplementation.

| Methods | Random | | Popular | | Adversarial | | Average | |
|---|---|---|---|---|---|---|---|---|
| | Acc↑ | F1↑ | Acc↑ | F1↑ | Acc↑ | F1↑ | Acc↑ | F1↑ |
| LLaVA-1.5-7B | 88.20 | 87.40 | 86.10 | 85.50 | 82.30 | 82.10 | 85.53 | 85.00 |
| + VCD (Leng et al., 2024) | 88.50 | 87.60 | 86.30 | 85.80 | 82.30 | 82.40 | 85.70 | 85.27 |
| + AGLA (An et al., 2025) | 87.73 | 86.35 | 86.47 | 85.15 | 84.40 | 83.26 | 86.20 | 84.92 |
| + VAF (Yin et al., 2025) | 89.80 | 89.40 | 87.50 | 87.40 | 83.40 | 82.60 | 86.90 | 86.47 |
| + MemVR (Zou et al., 2024) | 88.50 | 87.34 | 87.10 | 86.01 | **85.20** | 84.28 | 86.93 | 85.88 |
| + Ours | **89.93** | **89.43** | **88.33** | **87.95** | 84.73 | **84.79** | **87.66** | **87.39** |
| LLaVA-NeXT-7B | 88.83 | 87.60 | 87.83 | 86.63 | 86.36 | 85.26 | 87.67 | 86.50 |
| + VCD* (Leng et al., 2024) | 82.03 | 78.33 | 81.53 | 77.88 | 79.77 | 76.04 | 81.11 | 77.42 |
| + AGLA* (An et al., 2025) | 80.06 | 75.25 | 79.97 | 75.16 | 79.37 | 74.60 | 79.80 | 75.00 |
| + VAF* (Yin et al., 2025) | 90.30 | 89.55 | 89.26 | 88.56 | **86.93** | 86.41 | 88.83 | 88.17 |
| + MemVR* (Zou et al., 2024) | 88.69 | 87.41 | 87.87 | 86.68 | 86.40 | 85.30 | 87.65 | 86.46 |
| + Ours | **91.26** | **90.82** | **89.97** | **89.6** | 86.53 | **86.52** | **89.25** | **88.98** |
| Qwen2.5-VL-7B | 88.87 | 87.60 | 87.83 | 86.61 | 86.77 | 85.60 | 87.82 | 86.60 |
| + VCD* (Leng et al., 2024) | 89.03 | 87.87 | 88.00 | 86.87 | 86.86 | 85.89 | 87.96 | 86.88 |
| + AGLA* (An et al., 2025) | 88.73 | 87.43 | 87.50 | 86.24 | 86.60 | 85.39 | 87.61 | 86.35 |
| + VAF* (Yin et al., 2025) | **90.30** | **89.40** | 89.06 | 88.25 | 87.36 | 86.66 | 88.91 | 88.10 |
| + MemVR* (Zou et al., 2024) | 89.30 | 88.15 | 88.17 | 87.06 | 86.90 | 85.87 | 88.12 | 87.03 |
| + Ours | 90.27 | 89.35 | **89.17** | **88.29** | **87.60** | **86.82** | **89.01** | **88.15** |

Table 2: Evaluation results on CHAIR and AMBER. $C_S$ and $C_I$ denote CHAIR$_S$ and CHAIR$_I$, respectively.

| Methods | CHAIR | | AMBER | | |
|---|---|---|---|---|---|
| | $C_S$↓ | $C_I$↓ | CHAIR↓ | Hal↓ | Cog↓ |
| LLaVA-1.5-7B | 51.0 | 14.0 | 7.6 | 35.4 | 4.2 |
| + VCD | 53.8 | 16.3 | 8.7 | 40.0 | 4.2 |
| + AGLA | 50.0 | 15.3 | 7.0 | 32.4 | 3.7 |
| + VAF | 52.6 | 14.2 | 7.8 | 35.1 | 4.1 |
| + MemVR | 46.6 | 13.0 | 8.0 | 37.7 | 4.4 |
| + Ours | **37.6** | **11.3** | **5.2** | **22.8** | **2.2** |

Table 3: Results on CHAIR and AMBER for LLaVA-NeXT and Qwen2.5-VL. Our method is also effective for other mainstream models.

| Methods | CHAIR | | AMBER | | |
|---|---|---|---|---|---|
| | $C_S$↓ | $C_I$↓ | CHAIR↓ | Hal↓ | Cog↓ |
| LLaVA-NeXT-7B | 30.6 | 8.7 | 8.0 | 46.6 | 4.0 |
| + Ours | **25.4** | **7.2** | **6.3** | **34.8** | **2.6** |
| Qwen2.5-VL-7B | 36.4 | 9.5 | 5.1 | 28.4 | 1.7 |
| + Ours | **29.8** | **8.0** | **4.7** | **22.7** | **1.3** |

## 4 EXPERIMENTS

### 4.1 EXPERIMENTAL SETUP

To evaluate the effectiveness of our proposed method, we conduct experiments across six benchmarks: POPE (Li et al., 2023), CHAIR (Rohrbach et al., 2018), AMBER (Wang et al., 2023a), MME (Fu et al., 2024), MM-Vet (Yu et al., 2023), and ScienceQA (Lu et al., 2022). To ensure reproducibility, all evaluations are conducted using the greedy decoding strategy. Unless otherwise stated, hyperparameters are fixed at $\alpha = 0.15$ and $\gamma = 0.25$. For a fair comparison, the value of $\beta$ and the range of layers are kept consistent with those used in the VAF (Yin et al., 2025) method. Additional details regarding datasets, evaluation metrics, and implementation configurations are provided in Appendix D.

Table 4: Components ablation experiments on the POPE and CHAIR benchmarks.

| Settings | POPE (Random) | | POPE (Popular) | | POPE (Adversarial) | | CHAIR | |
|---|---|---|---|---|---|---|---|---|
| | Acc↑ | F1↑ | Acc↑ | F1↑ | Acc↑ | F1↑ | $C_S$↓ | $C_I$↓ |
| LLaVA-1.5-7B | 88.20 | 87.40 | 86.10 | 85.50 | 82.30 | 82.10 | 51.0 | 14.0 |
| W/o SVE | 89.47 | 88.75 | 88.20 | 87.56 | **85.00** | 84.70 | 41.2 | 12.5 |
| W/o SCA | 89.13 | 88.30 | 87.53 | 86.80 | 84.90 | 84.45 | 49.0 | 13.6 |
| Our Full VisionFocus | **89.93** | **89.43** | **88.33** | **87.95** | 84.73 | **84.79** | **37.6** | **11.3** |

## 4.2 MAIN RESULTS

We conduct hallucination evaluations on both discriminative (POPE) and generative (CHAIR and AMBER) tasks, and we set the maximum generation length to 512 during the evaluation. In addition, we also assess the performance of our method on general-purpose benchmarks (see Appendix C.1).

**Results on Hallucination Benchmarks.** As presented in Tab. 1, for LLaVA-1.5-7B, VisionFocus outperforms the baseline and existing methods on POPE, with average gains of 2.5% in accuracy and 2.8% in F1-score across three settings. As shown in Tab. 2, compared with vanilla LLaVA-1.5-7B, VisionFocus achieves substantial improvements on CHAIR, reducing $C_S$ and $C_I$ by 26.3% and 19.3%, respectively, significantly surpassing existing methods. Moreover, VisionFocus achieves the best performance on the AMBER generative task, outperforming the baseline and other methods across various metrics.

**Results on LLaVA-NeXT and Qwen2.5-VL.** To demonstrate the model-agnostic applicability of VisionFocus, we evaluate it on LLaVA-NeXT-7B and Qwen2.5-VL-7B-Instruct. The detailed settings are provided in Appendix D.3. As shown in Tab. 1 and Tab. 3, VisionFocus consistently delivers substantial improvements on POPE across various MLLMs. For example, on LLaVA-NeXT-7B, VisionFocus achieves average improvements of 1.8% in accuracy and 2.9% in F1-score across three settings. In addition, VisionFocus also significantly reduces the hallucination rate on CHAIR and AMBER. These results underscore the robustness of VisionFocus across mainstream MLLMs.

## 4.3 ABLATION STUDIES

This section presents ablation studies on the core modules. Additional experiments on hyperparameters, decoding strategies, and case study are provided in Appendix C.4, C.2, and E, respectively.

**Components Ablation.** We conduct an ablation study on the POPE and CHAIR benchmarks to evaluate the contribution of each component in VisionFocus, as shown in Tab. 4. The results show that removing any component causes a performance decline, highlighting the importance of strengthening the model's focus on vital global and local visual details to boost its overall performance.

## 5 CONCLUDING REMARKS

**Summary.** In this work, we address the critical challenge of hallucinations in MLLMs. While prior methods have shown promise, they often incur significant inference latency or may even exacerbate hallucinations. To tackle these issues, we propose VisionFocus, a framework that guides the model during decoding to focus on informative visual tokens while avoiding the excessive amplification of irrelevant or distracting visual information. Extensive experiments across six benchmarks demonstrate the effectiveness of VisionFocus in mitigating hallucinations and improving general performance. Importantly, VisionFocus is a plug-and-play, task-agnostic, and efficient approach that requires no additional fine-tuning, highlighting its broad applicability.

**Limitation.** Currently, VisionFocus lacks the ability to incorporate spatiotemporal information, which may limit its effectiveness in handling hallucinations that require long-term reasoning in video-based MLLMs. This limitation underscores the need for further research in this area.

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

# VisionFocus: Towards Efficient Hallucination Mitigation via Token-Aware Visual Enhancement

## Supplementary Material

### OVERVIEW

This material provides supplementary details to the main paper, including the following sections:

- (A) Motivation Details
    - (A.1) Attention Distribution Visualization
    - (A.2) Experimental Setup for Visual Token Analysis
    - (A.3) Comparison of SCA and Q-K Attention
- (B) Method Details
    - (B.1) Implementation Details of $F(\mathbf{A}, \alpha)$
    - (B.2) Implementation Details of $\mathbf{A}_{\text{avg}}$
    - (B.3) Implementation Details of SCA
- (C) Additional Experiments
    - (C.1) Results on General-Purpose Benchmarks
    - (C.2) Effect of Sampling Strategies
    - (C.3) Comparison of Inference Speed
    - (C.4) Hyperparameter sensitivity analysis.
- (D) Evaluation Details
    - (D.1) Benchmarks and Metrics
    - (D.2) Backbones and Baselines
    - (D.3) Reproducibility
- (E) Case Study
- (F) The Use of Large Language Models

## A    MOTIVATION DETAILS

In this section, we elaborate on the observations presented in the main paper Sec. 2.3 and Sec. 3.3. We focus on three critical phenomena: (1) as the number of layers increases, attention from the [CLS] token to visual tokens (LLaVA-1.5, LLaVA-NeXT), as well as attention among visual tokens (SigLIP, Qwen2.5-VL), gradually converges on a small subset of tokens, as illustrated in the main paper Fig. 2(a), (2) not all visual tokens contribute to mitigating hallucinations, and amplifying attention to some tokens may even worsen hallucinations, shown in the main paper Fig. 2(b), and (3) SCA outperforms the standard Q-K attention mechanism in capturing critical local visual information, as shown in the main paper Fig. 4.

### A.1    ATTENTION DISTRIBUTION VISUALIZATION

**Attention Distribution Across Layers.** We analyze the attention distribution across different layers of CLIP (Radford et al., 2021) within LLaVA-1.5-7B (Liu et al., 2024a). As illustrated in Fig. 5, attention in shallower layers (e.g., layer 1 and layer 5) is relatively dispersed across visual tokens. However, as layer depth increases, the attention progressively concentrates on a smaller subset of tokens. These highly attended tokens typically aggregate substantial information from all visual tokens and contain global visual information about the image (Yang et al., 2025), which plays a critical role in mitigating hallucinations.

**Attention Aggregation in SigLIP and LLaVA-NeXT.** To verify the universality of the phenomenon in which deep-layer attention concentrates on a small number of visual tokens, we further visualize the attention distributions of various types of visual encoders.

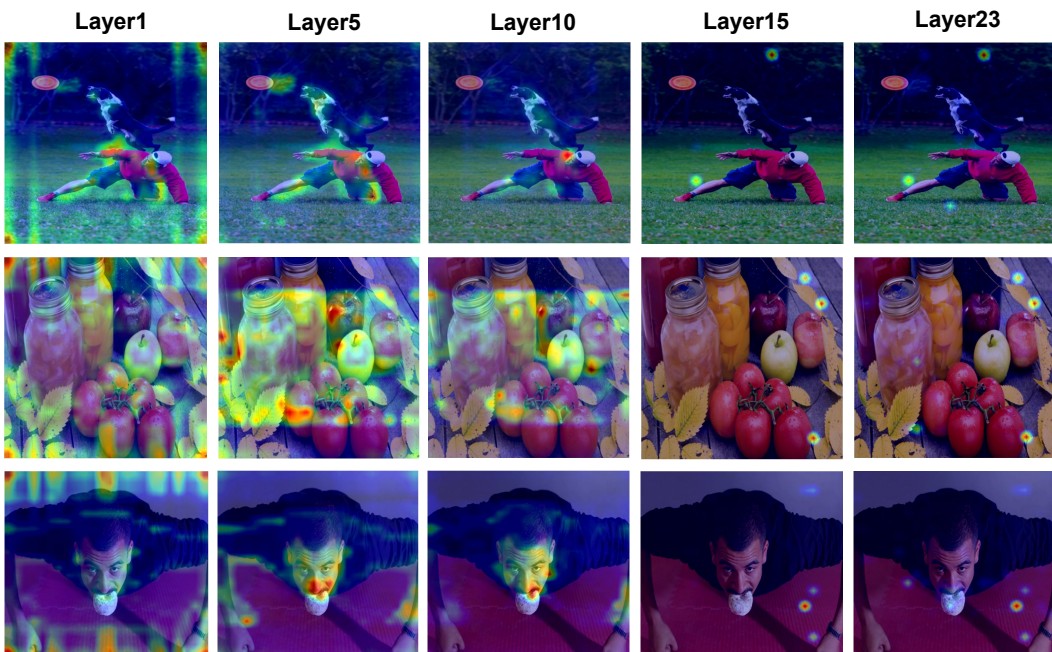

Figure 5: **Attention distribution across layers.** We visualize the attention from the [CLS] token to visual tokens using LLaVA-1.5. As the layer depth increases, the attention becomes increasingly concentrated on a limited subset of tokens, which typically aggregate global visual information.

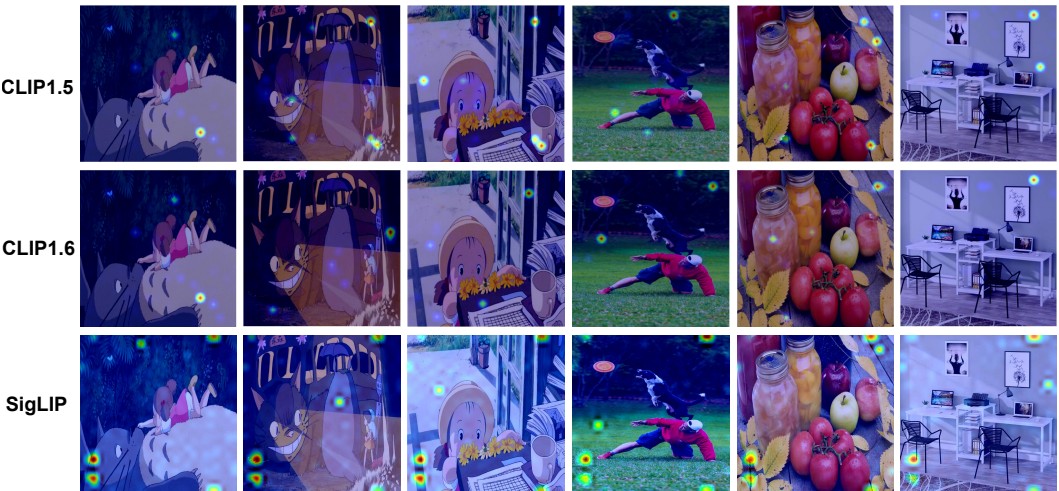

Figure 6: **Attention distributions in the penultimate layer of different vision encoders.** CLIP1.5 denotes the vision encoder used in LLaVA-1.5, and CLIP1.6 denotes the vision encoder used in LLaVA-NeXT. The results indicate that attention in the deeper layers consistently concentrates on a small subset of visual tokens across different vision encoders.

Specifically, we present the attention patterns in the penultimate layer of SigLIP (Zhai et al., 2023) and LLaVA-NeXT (Liu et al., 2024b). For LLaVA-NeXT, we visualize the attention from the [CLS] token to other visual tokens. For SigLIP, in the same layer, we show the average attention each visual token receives from all other tokens in the sequence (see Appendix B.2 for details). As shown in Fig. 6, the results indicate that, across different visual encoders, attention in the penultimate layer generally concentrates on a small subset of tokens.

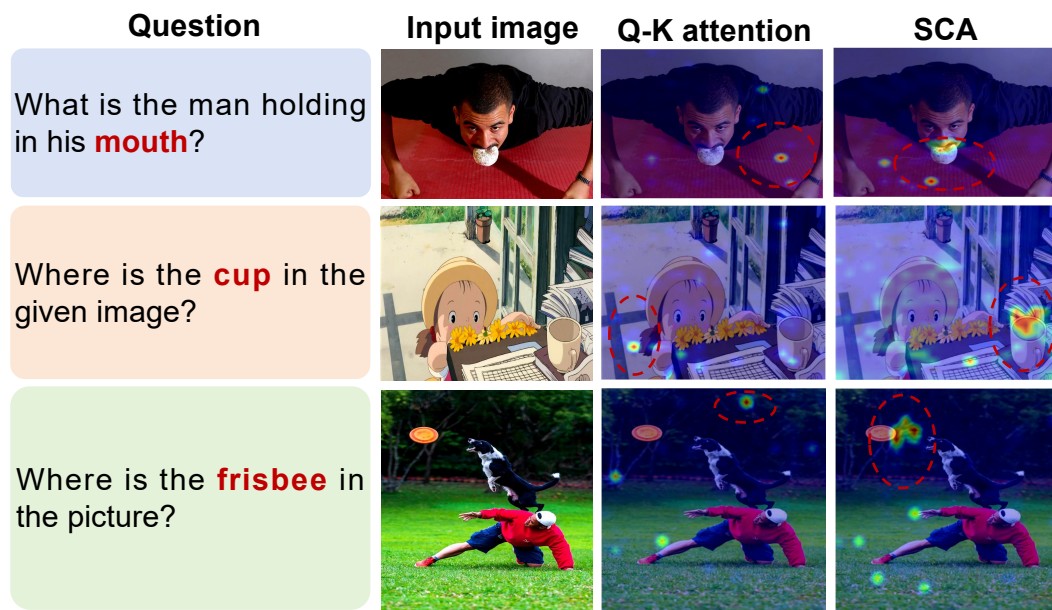

Figure 7: **Comparison of SCA and Q-K attention in localizing relevant local visual cues.** We visualize the attention weights over visual inputs for both attention mechanisms when the model generates key tokens (highlighted in **red bold**).

## A.2 EXPERIMENTAL SETUP FOR VISUAL TOKEN ANALYSIS

In this section, we describe the implementation details of the analysis presented in the main paper Sec. 2.3, which investigates the relationship between the attention levels of visual tokens and their impact on mitigating hallucinations.

Concretely, we rank the visual tokens according to the attention scores assigned by the [CLS] token in the penultimate layer of the vision encoder, from highest to lowest. We then select the top and bottom 5%, 10%, and 20% of tokens for enhancement. During the inference stage of the LLM, we apply a scaling factor of 1.3 to the attention scores of the selected visual tokens, in accordance with the strategy used in the VAF (Yin et al., 2025) method. To further illustrate the differential contributions of visual tokens, we introduce two additional conditions. One is a random selection strategy (Random), and the other is the vanilla LLaVA-1.5 model without any modification (Baseline). The results imply that not all visual tokens contribute positively to the model's reasoning, and some may even introduce negative effects. This highlights the need for a selective visual enhancement strategy to effectively mitigate hallucinations.

## A.3 COMPARISON OF SCA AND Q-K ATTENTION

To better demonstrate that SCA outperforms standard Q-K attention in identifying critical local information, we provide additional visualizations of attention distributions for both mechanisms using more examples in LLaVA-1.5.

As shown in Fig. 7, we present the model's attention weights over the visual inputs when generating key tokens, which are highlighted in **red bold**. Results indicate that SCA consistently demonstrates a stronger ability to focus on critical local visual details compared to the standard Q-K attention, which highlights that incorporating SCA significantly improves overall visual grounding in MLLMs.

## B METHOD DETAILS

In this section, we present the implementation details of our method. Appendix B.1 describes how we assign differential enhancement weights to visual tokens. Appendix B.2 presents how our method

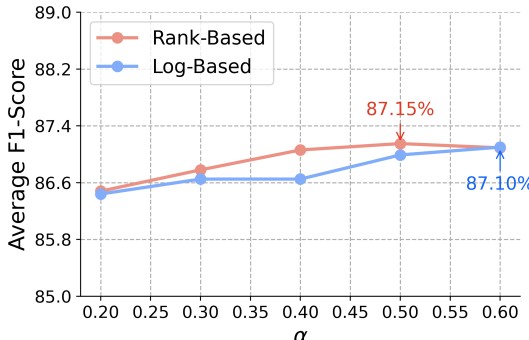

Figure 8: **Impact of different enhancement weights allocation strategies on POPE.** $\alpha$ denotes the predefined enhancement range, and a higher average F1-Score represents better performance.

applies to visual encoders that do not include a [CLS] token. Appendix B.3 discusses the integration of SCA into MLLMs and introduces several variants of the SCA mechanism.

### B.1 IMPLEMENTATION DETAILS OF $F(\mathbf{A}, \alpha)$

To examine the impact of different strategies for distributing enhancement weights under a given attention matrix and enhancement range $\alpha$, we test two approaches: Rank-Based and Log-Based.

Specifically, the Rank-Based strategy assigns enhancement weights in a linearly decreasing manner according to the descending order of attention scores. In contrast, the Log-Based strategy applies a logarithmic transformation to the attention scores, followed by normalization within the specified $\alpha$ range. We conduct comparative experiments using the LLaVA-1.5-7B on POPE, keeping the $\alpha$ value consistent across both strategies.

As illustrated in Fig. 8, the results indicate that the Rank-Based linear decay strategy slightly outperforms the Log-Based normalization strategy. This may be because there is substantial variance in attention scores across visual tokens. Although the Log-Based method alleviates this variance through logarithmic transformation and normalization, it may still underemphasize certain critical visual tokens during inference. In contrast, the Rank-Based approach more effectively prioritizes these key tokens. Therefore, we adopt the Rank-Based scheme as the enhancement weight assignment strategy in the subsequent experiments.

### B.2 IMPLEMENTATION DETAILS OF $\mathbf{A}_{\text{AVG}}$

Although most mainstream vision encoders, such as CLIP (Radford et al., 2021), OpenCLIP (Cherti et al., 2023), and LanguageBind (Zhu et al., 2023a), use the [CLS] token to aggregate information, SigLIP (Zhai et al., 2023) and the vision encoder of Qwen2.5-VL (Bai et al., 2025) do not include the [CLS] token. To demonstrate the generalizability of the proposed VisionFocus, this section describes its application to vision encoders that do not incorporate the [CLS] token.

Specifically, let $\mathbf{Q} \in \mathbb{R}^{H \times N \times D}$ represent the query embeddings in the penultimate layer of the vision encoder, where $H$ and $N$ denote the total number of attention heads and visual tokens, respectively. $\mathbf{K} \in \mathbb{R}^{H \times N \times D}$ corresponds to the key embeddings in the same layer. We first calculate the attention score $\mathbf{E} \in \mathbb{R}^{H \times N \times N}$ as follows:

$$\mathbf{E} = \text{SoftMax}\left(\frac{\mathbf{Q}\mathbf{K}^{\top}}{\sqrt{d}}\right), \tag{11}$$

where $d$ represents the dimension of each attention head. By averaging across the head dimension, we obtain an aggregated attention matrix $\hat{\mathbf{E}} \in \mathbb{R}^{N \times N}$. To identify key visual tokens, we calculate the average attention each token receives from all others in the sequence, denoted as $\mathbf{A}_{\text{avg}}$. Specifically, we compute the average of $\hat{\mathbf{E}}$ along dim $= 0$, which reflects how much each token is attended to by others, representing its importance.

## B.3 IMPLEMENTATION DETAILS OF SCA

**Integration Strategies.** We investigate the scope and position of integrating the SCA mechanism into MLLMs using LLaVA-1.5 on the POPE benchmark (Li et al., 2023).

Specifically, we investigate two settings regarding the scope of SCA application, with the key-value cache enabled by default. In Setting 1, during the prefill stage, SCA is applied exclusively to attention computations among visual tokens. During the decoding stage, SCA is applied only to attention calculations from the predicted token to the visual tokens. In Setting 2, SCA is applied to all attention computations throughout both the prefill and decoding stages. In addition, for the integration position, we explore three different fusion strategies as follows:

**Fusion1:**

$$\mathbf{Q} = \text{Proj}_q(\mathbf{X}), \mathbf{K} = \text{Proj}_k(\mathbf{X}), \mathbf{V} = \text{Proj}_v(\mathbf{X}), \tag{12}$$

$$\mathbf{Y} = \mathbf{X} + \text{Proj}(\text{SoftMax}(\hat{\mathbf{Z}}) \cdot \mathbf{V} + \gamma \cdot \text{Attn}_{\text{SCA}} \cdot \mathbf{V}). \tag{13}$$

**Fusion2:**

$$\mathbf{Y} = \mathbf{X} + \text{Proj}\left(\text{SoftMax}\left(\hat{\mathbf{Z}} + \gamma \cdot \left(\frac{\mathbf{K}\mathbf{K}^{\mathsf{T}}}{\sqrt{d}}\right)\right) \cdot \mathbf{V}\right). \tag{14}$$

**Fusion3:**

$$\mathbf{Y} = \mathbf{X} + \text{Proj}(\text{SoftMax}(\hat{\mathbf{Z}}) \cdot \mathbf{V}), \tag{15}$$

$$\mathbf{O} = \mathbf{Y} + \text{FFN}(\mathbf{Y}), \tag{16}$$

$$\hat{\mathbf{O}} = \mathbf{O} + \gamma \cdot \text{Attn}_{\text{SCA}} \cdot \mathbf{V}, \tag{17}$$

where $\mathbf{Q}$, $\mathbf{K}$, and $\mathbf{V}$ denote the query, key, and value embeddings of the LLM, respectively, and $\hat{\mathbf{Z}}$ represents the attention matrix modified by the SVE module described in the main paper. Proj denotes projection layers, and FFN denotes a feed-forward network. For simplicity, normalization operations are omitted.

As shown in Tab. 5, the results indicate that the combination of Setting 1 and Fusion 1 achieves the best overall performance across three settings on the POPE benchmark. Therefore, we adopt Setting 1 and Fusion 1 as the default strategy for integrating SCA with MLLMs. Moreover, due to architectural differences between Qwen2.5-VL and LLaVA-1.5, we observe that applying SCA during the prefill stage to compute attention scores among visual tokens leads to suboptimal results on Qwen2.5-VL. Therefore, for Qwen2.5-VL, we apply SCA only during the decoding stage and leave the prefill stage unchanged.

**Further Explorations of SCA.** In this section, we further examine the reasons behind the effectiveness of SCA. In the standard Q-K attention mechanism, attention scores are computed from representations generated by different projection matrices. Since each projection matrix is functionally distinct, this can hinder the model's ability to focus on the semantic content most relevant to its current representations. In contrast, computing attention scores using representations from the same projection matrix eliminates these functional discrepancies, allowing the model to more effectively capture the semantics embedded in its own representations.

Building on this insight, we explore three variants of Attn$_{\text{SCA}}$: Q-Q, K-K, and V-V, where the attention scores are computed using representations derived from the same projection matrix, respectively. Following the optimal integration strategy described above, all experiments are conducted with $\gamma$ fixed within the range of 0 to 0.5. We report the best performance achieved by each variant on POPE and compare the results with those of the standard Q-K attention.

As shown in Tab. 6, all variants outperform the baseline. Among them, the V-V variant achieves the best performance across multiple metrics. This may be because computing Attn$_{\text{SCA}}$ based on value states can better capture the semantic relationships among the value embeddings, which are directly involved in output construction within the attention mechanism. Therefore, we adopt the V-V variant as the default method for calculating Attn$_{\text{SCA}}$.

Table 5: Impact of different SCA integration strategies on POPE benchmark. Baseline denotes the vanilla LLaVA-1.5-7B, and the best results are highlighted in **bold**.

| Model | Settings | Random | | Popular | | Adversarial | |
|---|---|---|---|---|---|---|---|
| | | Acc ↑ | F1 ↑ | Acc ↑ | F1 ↑ | Acc ↑ | F1 ↑ |
| LLaVA-1.5-7B | Baseline | 88.20 | 87.40 | 86.10 | 85.50 | 82.30 | 82.10 |
| | Setting 1, Fusion 1 | **89.63** | **88.94** | **88.27** | **87.67** | 84.80 | **84.58** |
| | Setting 2, Fusion 1 | 88.17 | 88.32 | 86.00 | 86.48 | 78.80 | 80.85 |
| | Setting 1, Fusion 2 | 88.83 | 87.78 | 87.67 | 86.67 | **85.23** | 84.45 |
| | Setting 1, Fusion 3 | 89.17 | 88.20 | 87.97 | 87.07 | 85.17 | 84.52 |

Table 6: Performance of different SCA variants on the POPE benchmark. The Q-K (baseline) denotes the vanilla LLaVA-1.5.

| Model | Settings | Random | | Popular | | Adversarial | | Average | |
|---|---|---|---|---|---|---|---|---|---|
| | | Acc ↑ | F1 ↑ | Acc ↑ | F1 ↑ | Acc ↑ | F1 ↑ | Acc ↑ | F1 ↑ |
| LLaVA-1.5-7B | Q-K | 88.20 | 87.40 | 86.10 | 85.50 | 82.30 | 82.10 | 85.53 | 85.00 |
| | K-K | 89.47 | 88.70 | 87.87 | 87.20 | **85.07** | 84.70 | 87.47 | 86.87 |
| | V-V | **89.93** | **89.43** | **88.33** | **87.95** | 84.73 | **84.79** | **87.66** | **87.39** |
| | Q-Q | 89.47 | 88.71 | 87.73 | 87.09 | 84.77 | 84.45 | 87.32 | 86.75 |

Table 7: Performance evaluation on the MME Hallucination subset, MM-Vet, and ScienceQA.

| Methods | MME-Hall | Object-Level | | Attribute-Level | | MM-Vet | ScienceQA | |
|---|---|---|---|---|---|---|---|---|
| | Total ↑ | Existence ↑ | Count ↑ | Position ↑ | Color ↑ | Acc ↑ | Image Acc ↑ | Acc ↑ |
| LLaVA-1.5-7B | 610 | 185 | 146.67 | 128.33 | 150 | 28.37 | 66.80 | 68.00 |
| + Ours | **625** | **185** | **155.00** | **130.00** | **155** | **29.70** | **68.27** | **69.56** |

# C    ADDITIONAL EXPERIMENTS

In this section, we present additional evaluations and ablation studies to further validate the effectiveness of our proposed method.

## C.1    RESULTS ON GENERAL-PURPOSE BENCHMARKS

We further evaluate VisionFocus on three general-purpose benchmarks, including MME (Fu et al., 2024), MM-Vet (Yu et al., 2023), and ScienceQA (Lu et al., 2022), to examine its robustness beyond hallucination-specific scenarios. As shown in Tab. 7, in the hallucination-related subset of MME, VisionFocus achieves an improvement of 15 points in both object-level and attribute-level hallucination metrics. It also outperforms the baseline on MM-Vet and ScienceQA across various metrics. These results demonstrate that VisionFocus effectively mitigates hallucination while preserving strong overall performance on general benchmarks.

## C.2    EFFECT OF SAMPLING STRATEGIES

We evaluate the effectiveness of the VisionFocus method in mitigating hallucinations under various sampling strategies using LLaVA-1.5-7B on the COCO-Random subset of the POPE benchmark.

Specifically, we examine six sampling strategies: Top-P sampling ($p = 0.7$), Top-K sampling ($k = 50$), direct sampling ($temp = 1.0$), Top-P sampling with temperature ($p = 0.7$, $temp = 0.5$), Top-K sampling with temperature ($k = 50$, $temp = 0.5$), and greedy decoding. As shown in Tab. 8,

Table 8: Results of VisionFocus under different sampling strategies on the COCO-Random subset of POPE using LLaVA-1.5. Regular denotes the vanilla LLaVA-1.5.

| Sampling Strategy | Methods | Accuracy ↑ | Precision | Recall | F1-Score ↑ |
|---|---|---|---|---|---|
| Greedy | Regular | 88.20 | 94.40 | 81.40 | 87.40 |
| | VisionFocus | **89.93** (+1.73) | 94.17 | 85.13 | **89.43** (+2.03) |
| Direct Sampling | Regular | 84.17 | 92.74 | 74.13 | 82.40 |
| | VisionFocus | **85.53** (+1.40) | 88.68 | 81.47 | **84.92** (+2.52) |
| Top P | Regular | 87.27 | 96.43 | 77.40 | 85.87 |
| | VisionFocus | **89.27** (+2.00) | 93.63 | 84.27 | **88.70** (+2.83) |
| Top K | Regular | 83.97 | 92.85 | 73.60 | 82.11 |
| | VisionFocus | **85.83** (+1.86) | 89.55 | 81.13 | **85.13** (+3.02) |
| Top P + Temp = 0.5 | Regular | 88.13 | 97.12 | 78.60 | 86.88 |
| | VisionFocus | **89.97** (+1.84) | 94.51 | 84.87 | **89.43** (+2.55) |
| Top K + Temp = 0.5 | Regular | 87.83 | 96.94 | 78.13 | 86.53 |
| | VisionFocus | **89.06** (+1.23) | 92.65 | 84.87 | **88.59** (+2.06) |

Table 9: Comparison of decoding speed across different methods on LLaVA-1.5. The TPS refers to Tokens Per Second, and the Average Latency represents the relative decoding delay compared to the baseline (Greedy).

| Methods | Average TPS ↑ | Average Latency ↓ |
|---|---|---|
| Greedy | 33.42 | 1.000× |
| VCD | 16.90 | 1.978× |
| AGLA | 16.50 | 2.025× |
| VAF | 33.04 | 1.012× |
| MemVR | 19.40 | 1.723× |
| VisionFocus (ours) | 32.95 | 1.013× |

VisionFocus consistently reduces hallucinations across all sampling strategies, demonstrating its robustness and general applicability.

### C.3  COMPARISON OF INFERENCE SPEED

We randomly sample three images from the CHAIR dataset and prompt the model (LLaVA-1.5-7B) with the instruction "Please describe the image in detail." We set the maximum output length to 512 tokens and measure the average number of tokens generated per second across three images for each method. Additionally, we also compute the average decoding latency of each method relative to the baseline (Greedy).

As shown in Tab. 9, our method introduces only a modest 1.3% increase in latency relative to the baseline, indicating minimal impact on decoding efficiency. In contrast, Contrastive Decoding (CD) methods nearly double the decoding speed due to their two-stage inference pipeline. These results underscore the practical advantages of our approach, which achieves effective hallucination mitigation while maintaining competitive decoding speed. This efficiency makes it particularly suitable for real-world applications where low-latency responses are critical.

### C.4  HYPERPARAMETER SENSITIVITY ANALYSIS.

We conduct a sensitivity analysis to examine the effect of hyperparameters $\alpha$ and $\gamma$ on model performance, measured by average Accuracy and F1-score on the POPE benchmark. The results are

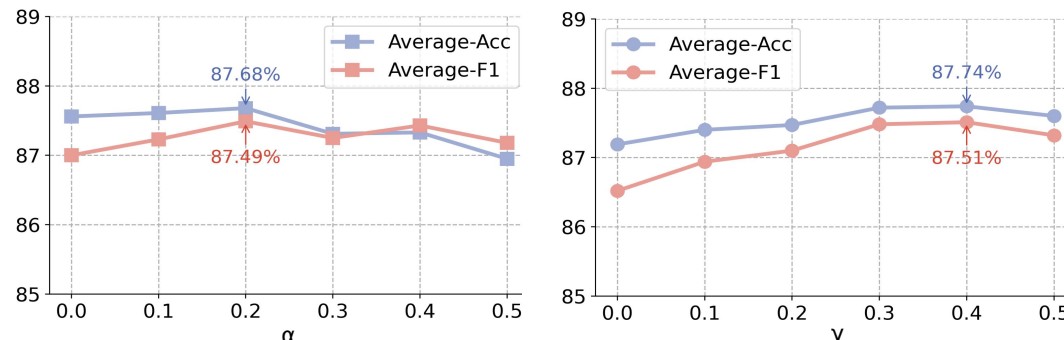

Figure 9: Ablation study of $\alpha$ and $\gamma$ on POPE. Average-Acc and Average-F1 denote the mean accuracy and F1-score across three evaluation settings.

depicted in Fig. 9. For $\alpha$, we observe that hallucination rates are significantly reduced when $\alpha$ falls within the range of (0.0, 0.2). However, setting $\alpha > 0.2$ leads to a notable decline in performance. For $\gamma$, performance consistently improves as $\gamma$ increases from 0.0 to 0.4, with the most substantial gains observed in the interval [0.2, 0.3]. When $\gamma$ exceeds 0.5, performance begins to deteriorate. We speculate that the degradation is caused by the model overemphasizing visual information, resulting in insufficient attention to the user's query. Overall, VisionFocus is insensitive to hyperparameter variations, demonstrating its strong robustness.

## D EVALUATION DETAILS

In this section, we provide detailed information about the evaluation process. The evaluation benchmarks we used are described in Appendix D.1. In Appendix D.2, we outline the baseline methods used for comparison. In Appendix D.3, we present the detailed evaluation setup.

### D.1 BENCHMARKS AND METRICS

**Datasets.** To evaluate the effectiveness of our proposed method, we perform extensive experiments on six benchmarks. These benchmarks include three specifically designed for hallucination evaluation and three general-purpose ones for assessing overall model performance. In this section, we provide additional details on the benchmarks referenced in the main paper.

**POPE** (Li et al., 2023) is a Visual Question Answering (VQA)-based benchmark that evaluates whether Multimodal Large Language Models (MLLMs) exhibit object hallucination. The evaluation consists of questions in the format "Is there a [object] in the image?", followed by the prompt "Answer the question using a single word or phrase" to enforce a binary response format. Three sampling strategies are used to select objects: random sampling, popular sampling, and adversarial sampling. Evaluation metrics include Accuracy, Precision, Recall, and F1-score.

**CHAIR** (Rohrbach et al., 2018) measures object hallucination in image captions by comparing the objects mentioned in generated captions with those present in the ground-truth annotations. Following previous studies (Zou et al., 2024; Yue et al., 2024), we randomly sample 500 images from the MSCOCO dataset and adopt the official $CHAIR_S$ and $CHAIR_I$ scores for evaluation.

**AMBER** (Wang et al., 2023a) is a multi-dimensional, LLM-free benchmark designed to evaluate hallucination in both generative and discriminative tasks. It considers various types of hallucinations, including object existence, object attributes, and inter-object relations. For generative tasks, evaluation metrics include CHAIR, Cover, Hal, and Cog. For discriminative tasks, performance is measured using Accuracy, Precision, Recall, and F1-score. The final assessment is based on the aggregated AMBER Score, which provides a comprehensive measure of overall performance.

**MME** (Fu et al., 2024) evaluates MLLMs across two task types: perceptual and cognitive. Perceptual tasks assess fine-grained visual understanding, including object presence, counting, spatial position, color, celebrity and poster recognition, scene and landmark classification, artwork identifi-

cation, and OCR. Cognitive tasks cover commonsense reasoning, numerical calculation, translation, and code reasoning. All questions are formatted as binary answers (yes/no).

**MM-Vet** (Yu et al., 2023) is a comprehensive benchmark that assesses six core MLLMs capabilities: recognition, OCR, knowledge-based question answering, language generation, spatial reasoning, and mathematical computation. It introduces an open-ended response evaluation framework based on Large Language Models (LLMs), enabling flexible question and answer formats while providing standardized scoring.

**ScienceQA** (Lu et al., 2022) is a multiple-choice benchmark designed to evaluate zero-shot generalization in scientific question answering. It includes multimodal questions spanning a wide range of science topics, with annotated answers supported by corresponding lectures and explanations. These annotations offer both general external knowledge and specific reasoning required to derive the correct answer. In our study, we evaluate model performance on the image subset of ScienceQA to assess its capability in multimodal scientific reasoning.

## D.2 BACKBONES AND BASELINES

To demonstrate the robustness of our method, we adopt three widely used MLLMs: LLaVA-1.5 (Liu et al., 2024a), LLaVA-NeXT (Liu et al., 2024b), and Qwen2.5-VL (Bai et al., 2025). In addition, we compare our approach with several training-free state-of-the-art methods for mitigating hallucinations. The details of these methods are as follows:

**VCD** (Leng et al., 2024) is a training-free method designed to mitigate hallucinations in MLLMs by enhancing their focus on image content. It achieves this by contrasting output distributions derived from both original and distorted visual inputs. This contrastive approach helps the model better align its responses with actual image content rather than relying on spurious correlations. The computational cost of a single inference step using VCD is approximately twice that of standard greedy decoding.

**AGLA** (An et al., 2025) is a training-free, plug-and-play method aimed at reducing hallucinations in MLLMs by enhancing attention to prompt-relevant visual features. It assembles global features for response generation and local features for visual discrimination. AGLA introduces an image-prompt matching scheme to extract local features relevant to the input prompt, creating an augmented image view that suppresses irrelevant content. By calibrating the output logit distribution using both the original (global) and augmented (local) features, AGLA improves visual grounding and reduces hallucinations. However, the computational cost of a single inference step using AGLA is also approximately twice that of standard greedy decoding.

**MemVR** (Zou et al., 2024) reduces hallucinations in MLLMs by reinforcing the model's utilization of visual information. Inspired by the human behavior of revisiting visual input when uncertain, MemVR introduces a "look-twice" mechanism that reinjects visual tokens as key-value memory into the model's Feed Forward Network at a mid-layer during inference. This reinjection is dynamically triggered based on model uncertainty.

**VAF** (Yin et al., 2025) identifies that hallucinations stem from insufficient attention to visual information during the decoding process. To address this, it proposes uniformly enhancing attention to all visual tokens while penalizing excessively high attention on system tokens. However, uniformly increasing attention to all visual tokens may exacerbate hallucinations.

## D.3 REPRODUCIBILITY

**Implementation Details.** We employ greedy search as the default decoding strategy across all benchmark experiments. For the hallucination benchmarks (POPE, CHAIR, and AMBER) and general-purpose benchmarks (MME, MM-Vet, and ScienceQA), we use the annotated questions as prompts and format them according to the input prompt templates specific to each multimodal large language model.

In LLaVA-NeXT-7B, the importance of visual tokens is determined using the [CLS] token, while in Qwen2.5-VL-7B (without the [CLS] token), it is estimated based on the average attention each visual token receives from the remaining visual tokens.

The POPE benchmark is evaluated using the COCO dataset. MM-Vet is assessed via its official online evaluator, which relies on GPT-4 for scoring. CHAIR is evaluated on a randomly sampled subset of 500 images from the COCO Val 2014 dataset, selected by Less is More (Yue et al., 2024). We adopt the original dataset released by Less is More without any modification and use a unified prompt: "Please describe the image in detail."

All VisionFocus evaluations are conducted using greedy decoding with the following settings: do_sample=False, temperature=0, $\alpha = 0.15$, and $\gamma = 0.25$. To ensure a fair comparison with the VAF method (Yin et al., 2025), we adopt the same configuration by setting $\beta = 0.85$ and restricting the layer range to $8 < \text{Layers} < 15$. For VCD, AGLA, MemVR, and VAF, we standardize the decoding strategy to greedy search while keeping all other hyperparameters consistent with those reported in the original papers or provided in their official code repositories.

**Experimental Code.** To ensure transparency and reproducibility, all code, datasets, and detailed tutorials will be made publicly available soon.

## E   CASE STUDY

We analyze specific cases for the discriminative task (Fig. 10 and Fig. 11), VQA task (Fig. 12), and generative task (Fig. 13, Fig. 14, Fig. 15, and Fig. 16), and compare the performance of different methods and models.

**Discriminative Task.** As shown in Fig. 10, VisionFocus exhibits a more focused and reasonable attention distribution over critical local regions than vanilla LLaVA-1.5-7B, while effectively mitigating hallucinations in MLLMs. As illustrated in Fig. 11, vanilla LLaVA-1.5-7B, VCD, and AGLA exhibit varying degrees of hallucination on the POPE benchmark, whereas VisionFocus demonstrates greater robustness and accuracy, thereby validating the effectiveness of our approach.

**VQA Task.** As shown in Fig. 12, in VQA tasks involving numerical reasoning and color recognition, VisionFocus surpasses vanilla LLaVA-1.5-7B and VAF in both reasoning accuracy and detail capture, indicating its stronger ability to perceive fine-grained features.

**Generative Task.** Fig. 13, Fig. 14, Fig. 15, and Fig. 16 present comparative results on generative tasks, where VisionFocus produces descriptions that are more consistent with image content, more coherent, and richer in detail than those of baseline models, further demonstrating its superiority in cross-modal understanding and generation.

## F   THE USE OF LARGE LANGUAGE MODELS

In this work, large language models (LLMs) are used exclusively for polishing the writing and checking grammar. They are not involved in research ideation, experimental design, data analysis, or the formulation of conclusions. All substantive intellectual contributions are made by the authors.

| **Image Input** | **Attention** (LLaVA-1.5-7b) | **Attention** (VisionFocus) |

Is there a **spoon** in the image?\nAnswer the question using a single word or phrase.

(hallucinated answer of **LLaVA-1.5-7b**) No.

Is there a **spoon** in the image?\nAnswer the question using a single word or phrase.

(correct answer given by **VisionFocus**) Yes.

Is there a **knife** in the image?\nAnswer the question using a single word or phrase.

(hallucinated answer of **LLaVA-1.5-7b**) No.

Is there a **knife** in the image?\nAnswer the question using a single word or phrase.

(correct answer given by **VisionFocus**) Yes.

Figure 10: **Comparison of hallucination and attention distribution between baseline and our VisionFocus.** The results indicate that our method achieves a more precise key attention distribution and a lower hallucination rate.

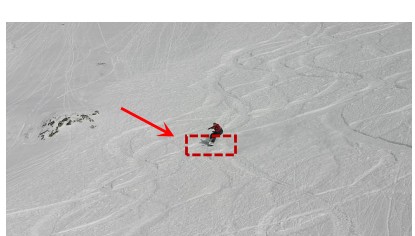
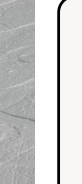

**Question:** Is there a **skis** in the image?\nAnswer the question using a single word or phrase.

**Ground Truth:** Yes

**LLaVA-1.5-7b Default:** No

**LLaVA-1.5-7b + VCD:** Yes

**LLaVA-1.5-7b + AGLA:** Yes

**LLaVA-1.5-7b + VisionFocus (ours):** Yes

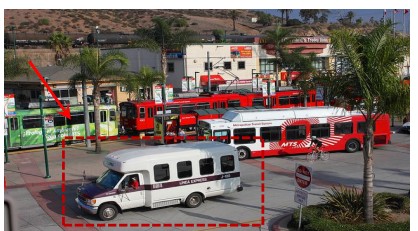

**Question:** Is there a **car** in the image?\nAnswer the question using a single word or phrase.

**Ground Truth:** Yes

**LLaVA-1.5-7b Default:** Yes

**LLaVA-1.5-7b + VCD:** No

**LLaVA-1.5-7b + AGLA:** No

**LLaVA-1.5-7b + VisionFocus (ours):** Yes

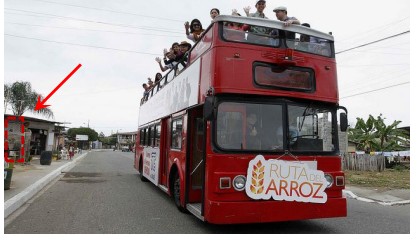

**Question:** Is there a **potted plant** in the image?\nAnswer the question using a single word or phrase.

**Ground Truth:** Yes

**LLaVA-1.5-7b Default:** No

**LLaVA-1.5-7b + VCD:** No

**LLaVA-1.5-7b + AGLA:** No

**LLaVA-1.5-7b + VisionFocus (ours):** Yes

Figure 11: **An illustration of VisionFocus correcting hallucinations on the POPE benchmark, with a focus on discriminative tasks.** Correct outputs are highlighted in green, while incorrect ones are marked in red.

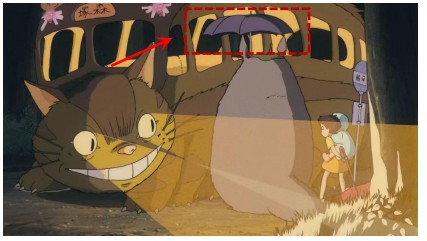

**Ground Truth:** Three

**LLaVA-1.5-7b Default: There are two apples in the image.**

**LLaVA-1.5-7b + VAF: There are two apples in the image.**

**LLaVA-1.5-7b + VisionFocus (ours):** There are three apples in the image.

**Question:** How many **apples** are there in the image?

**Ground Truth:** Purple

**LLaVA-1.5-7b Default:** The color of the umbrella in the given picture is blue.

**LLaVA-1.5-7b + VAF:** The color of the umbrella in the given picture is blue.

**LLaVA-1.5-7b + VisionFocus (ours):** The color of the umbrella in the given picture is purple.

**Question:** What's the color of the **umbrella** in the given picture?

Figure 12: **An illustration of VisionFocus mitigating hallucinations in randomly selected examples, focusing on numerical and color perception tasks.** Correct outputs are highlighted in green, while incorrect ones are marked in red.

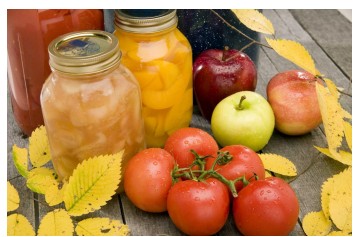

Please describe the given picture in detail.

**LLaVA-1.5-7b:**
The image features a wooden table with a variety of fruits and jams arranged on it. There are several apples, with some placed closer to the center of the table and others scattered around the edges. A few oranges are also present, with one located near the center and another towards the right side of the table.
In addition to the fruits, there are multiple jars of jam placed on the table. Some jars are positioned closer to the left side, while others are placed more towards the center and right side of the table. The combination of fruits and jams creates a visually appealing and inviting display.

**LLaVA-1.5-7b + VisionFocus (Ours):**
The image features a table filled with a variety of fruits and jars. There are several apples scattered across the table, with some placed closer to the front and others towards the back.
In addition to the fruits, there are multiple jars of different sizes and shapes placed on the table. The jars are arranged in various positions, with some closer to the front and others towards the back. The combination of fruits and jars creates a visually appealing and colorful display.

Figure 13: **An illustration of VisionFocus mitigating hallucinations in randomly selected examples, focusing on generative tasks.** Correct outputs are highlighted in green, while incorrect ones are marked in red.

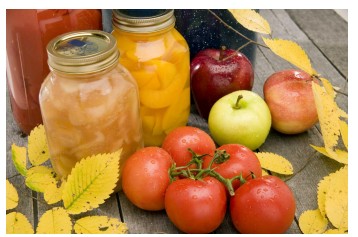

Please describe the given picture in detail.

**LLaVA-NeXT-7b:**
The image presents a vibrant still life arrangement on a wooden surface. At the center, there are three jars filled with a yellow liquid, possibly a fruit preserve or juice. To the left of these jars, there are two apples, one red and one green, with their stems visible. To the right of the jars, there are two more apples, one red and one green, with their stems also visible. In the foreground, there are four tomatoes, two red and two green, with their stems attached. The tomatoes are wet, suggesting they have been recently washed or are freshly picked. Scattered around the tomatoes are several yellow leaves, adding a touch of autumn to the scene.
The overall arrangement is aesthetically pleasing, with the colors of the fruits and vegetables contrasting nicely against the wooden background.

**LLaVA-NeXT-7b + VisionFocus (Ours):**
The image presents a vibrant display of fresh produce arranged on a wooden surface. Dominating the scene are several jars of preserves, their contents a mix of colors that suggest a variety of flavors. The jars are filled to the brim, their contents reflecting the light in a way that suggests they are freshly made. Scattered around the jars are several apples, their red and green hues adding a splash of color to the scene. They are fresh and shiny, their skins unblemished and their stems still attached. Adding to the natural beauty of the scene are fallen leaves, their colors muted and their edges curled up. They are scattered around the jars and apples, their presence adding a touch of autumn to the scene.
The overall image is one of abundance and freshness, a testament to the bounty of nature. It's a snapshot of a moment in time, a moment of stillness before the hustle and bustle of life resumes.

Figure 14: **An illustration of VisionFocus mitigating hallucinations in randomly selected examples, focusing on generative tasks.** Correct outputs are highlighted in green, while incorrect ones are marked in red.

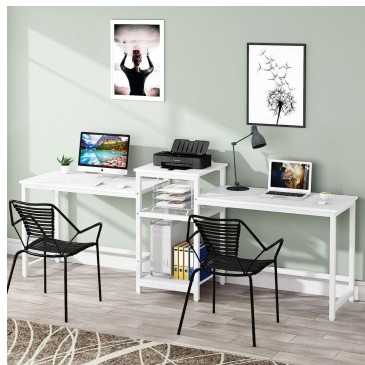

Please describe the given picture in detail.

**LLaVA-1.5-7b:**
The image features a well-organized home office with a white desk and two black chairs. On the desk, there is a laptop computer, a keyboard, and a mouse. A monitor is also present, positioned on the left side of the desk.
In addition to the computer setup, there are several books scattered around the desk, with some placed on the right side and others on the left. A potted plant can be seen in the background, adding a touch of greenery to the room. A clock is mounted on the wall, allowing the user to keep track of time while working.

**LLaVA-1.5-7b + VisionFocus (Ours):**
The image features a well-organized home office with a white desk and a laptop placed on it. The desk is equipped with a printer, a mouse, and a keyboard, making it a functional workspace. There are two chairs in the room, one on the left side and another on the right side of the desk.
In addition to the desk setup, there is a potted plants on the left side of the desk. Some books are also visible on the desk, adding to the organized appearance of the space.

Figure 15: **An illustration of VisionFocus mitigating hallucinations in randomly selected examples, focusing on generative tasks.** Correct outputs are highlighted in green, while incorrect ones are marked in red.

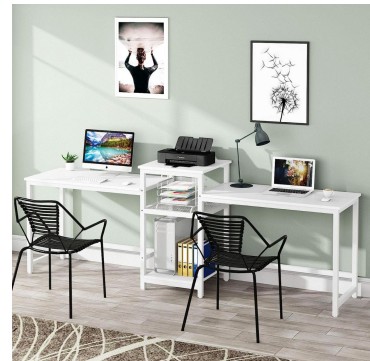



👤 Please describe the given picture in detail.



**LLaVA-NeXT-7b:**

The image depicts a modern, minimalist home office setup. The room has a light green wall, and the floor is covered with light-colored tiles. On the left side of the image, there is a white desk with a black computer monitor displaying a landscape image. Above the monitor, there is a black printer. To the right of the desk, there is a white shelf with a black lamp and a small plant. A black chair with a curved backrest is positioned in front of the desk.

On the right side of the image, there is another white desk with a laptop displaying a close-up of a flower. Above the laptop, there is a black picture frame with a black and white photograph of a person. To the right of the desk, there is a white shelf with a small stack of books and a white coffee cup. A black chair with a curved backrest is placed in front of this desk.

The room is well-lit, with natural light coming from the left side of the image, casting shadows on the wall and the floor. The overall style of the image is clean and modern, with a focus on functionality and simplicity. 🌋

**LLaVA-NeXT-7b + VisionFocus (Ours):**

The image depicts a modern, minimalist home office setup. The room has a light green wall with a framed picture of a person holding a surfboard. On the desk, there is a laptop displaying a scenic image of a beach with a rocky shoreline. To the right of the laptop, there is a black printer with a white paper tray. Above the desk, there is a black lamp with a gray shade. On the left side of the desk, there is a black chair with a woven seat and backrest. The floor is covered with light-colored tiles, and there is a patterned area rug with a white and gray color scheme.

The overall style of the image is clean and contemporary, with a focus on functionality and simplicity. 🕵️

Figure 16: **An illustration of VisionFocus mitigating hallucinations in randomly selected examples, focusing on generative tasks.** Correct outputs are highlighted in green, while incorrect ones are marked in red.

