# OpenReview forum: "VisionFocus: Towards Efficient Hallucination Mitigation via Token-Aware Visual Enhancement"
_ICLR.cc/2026/Conference — ICLR 2026 Conference Withdrawn Submission_

### Official Review · Reviewer_vMDs · 2025-10-17

**Soundness:** 3
**Presentation:** 3
**Contribution:** 2
**Rating:** 6
**Confidence:** 4

**Summary:**

This paper addresses hallucinations in multimodal LLMs and argues that prior training-free approaches either incur high latency via contrastive decoding or uniformly amplify all visual tokens, which can worsen hallucinations. The authors observe that attention within vision encoders concentrates on a restricted subset of tokens and that selectively enhancing these “vital” tokens is more effective than uniform boosting. They introduce VisionFocus, a plug-and-play, training-free method comprising (1) Selective Visual Enhancement (SVE)—implemented via Differential Visual Scaling (DVS) and Attention Sink Calibration (ASC) to up-weight informative visual tokens—and (2) Semantic Covariant Attention (SCA), which uses key states as queries so attention covaries with the current word, improving localization. The two modules are fused during LLM inference to balance global completeness and local precision.

## Motivation

This paper targets hallucination in MLLMs, noting that prior training-free methods either (i) use contrastive decoding (effective but slow due to multi-round inference) or (ii) uniformly boost attention to all visual tokens (fast but can worsen hallucinations). The authors observe attention in vision encoders concentrates on a restricted subset of tokens; selectively emphasizing these tokens—rather than uniform boosting—should better mitigate hallucinations.

## Methodology

They propose VisionFocus, a plug-and-play, training-free approach with two modules:

* Selective Visual Enhancement (SVE)—ranks visual tokens via encoder attention, then applies Differential Visual Scaling (DVS) to informative tokens and Attention Sink Calibration (ASC) to avoid over-attending system/instruction tokens.
* Semantic Covariant Attention (SCA)—replaces the standard query in Q-K attention with the key state so attention covaries with the current word, improving localization of fine-grained evidence.

## Experimental Results

Across six benchmarks (POPE, CHAIR, AMBER, MME, MM-Vet, ScienceQA) under greedy decoding, VisionFocus yields consistent gains: on LLaVA-1.5-7B, CHAIR metrics drop markedly and AMBER hallucination/cognition metrics improve, similar improvements hold for LLaVA-NeXT-7B and Qwen2.5-VL-7B. The method adds a little latency over baseline, much lower than CD-based baselines that roughly double decoding time.

## Analysis

Findings support the central claim that selective enhancement of high-value visual tokens reduces hallucinations more effectively than uniform boosting, and that making attention semantically covariant with the token being generated improves fine-grained grounding. The SVE and SCA work synergistically to deliver better accuracy with minimal computational overhead.

**Strengths:**

* 1. Training-free, plug-and-play, task-agnostic. VisionFocus requires no additional training or fine-tuning and is designed to be easily inserted at inference time, highlighting practical deployability.

* 2. Clear principle of selective token enhancement. The paper diagnoses why uniform attention boosting can worsen hallucinations and motivates selectively emphasizing high-value visual tokens, grounded in encoder attention analyses.

* 3. Well-motivated two-module design (SVE + SCA). SVE (with DVS/ASC) targets globally informative visual tokens, while SCA makes attention semantically covariant with the word being generated; their fusion is explicitly formulated.

* 4. Consistent gains on hallucination and general benchmarks. Substantial improvements on CHAIR/AMBER and POPE, with experiments spanning six benchmarks, including MME, MM-Vet, ScienceQA.

* 5. Cross-model generalization. The method improves LLaVA-1.5-7B and transfers to LLaVA-NeXT-7B and Qwen2.5-VL-7B with similar benefits, underscoring model-agnostic applicability.

* 6. Strong efficiency profile. Near-baseline decoding speed (1.013× latency), in contrast to CD-style baselines that nearly double latency.

* 7. Intuitive qualitative evidence. Visualizations show SCA produces semantically covariant attention maps that better localize evidence.

**Weaknesses:**

* 1. Limited evaluation benchmarks and models. It is recommended that the authors may test the performance on their proposed benchmarks (POPE, CHAIR, MME, MM-Vet, ScienceQA) for each of the baseline models (llava-v1.5-7b, Qwen2.5-VL-7B and llava-next-7B), and include the full evaluation scores in Appendix.

* 2. Hidden tuning sensitivity. Although the paper claims robustness, the sensitivity plot shows α > 0.2 and γ > 0.5 degrade performance—implying non-trivial tuning ranges and a trade-off between visual emphasis and instruction-following.

* 3. Limited architectural/scale diversity. Results center on 7B-class MLLMs (LLaVA-1.5-7B, LLaVA-NeXT-7B, Qwen2.5-VL-7B-Instruct). Evidence on larger scales is limited.

**Questions:**

* 1. Generalization beyond COCO/MM-Vet. How does VisionFocus perform on text-heavy or domain-specific images (documents, charts, UI, medical)? Any results on OCR-style or chart reasoning benchmarks?

* 2. Tuning protocol. Given that α and γ have effective ranges, how should practitioners pick them without validation labels? Any heuristic/auto-tuning scheme?

* 3. SCA integration details. Where is SCA injected, and what is the per-layer FLOPs/memory overhead?

* 4. Instruction-following trade-off. You speculate performance drops when γ is high due to over-emphasizing visual cues. Can you quantify impacts on instruction-following benchmarks or provide a dynamic schedule for γ that adapts to the prompt type?

* 5. why key state in the LLM contains the full semantic information of the associated token, as stated in Page 7 *Local Semantic Alignment*? Any further explanation or theoretical analysis on that?

* 6. Does it require adjusting the value of ratio gama when injecting related local visual semantics, for each different MLLM model and benchmark, to get the best result? If yes, please include the hyper-parameter settings for the evaluation results shown in the tables.

* 7. Since Key-Key pairs introduce more precise attention distributions than Query-Key pairs do, why increasing the value of ratio gama leads to downgraded performance in Figure 9?

* 8. Does the proposed approach only tends to be insensitive to hyperparameter variations for specific models and benchmarks? If no, more ablation study on different MLLM models on different benchmarks is preferred.

* 9. Case studies show a number of how VisionFocus fixes hallucinated content. But does VisionFocus always work? Any analysis on cases where VisionFocus fails to fix the hallucination, or even create hallucinated content while baseline model does not?

*More concerns please also refer to Weakness section. I would consider raising my score if authors could give reasonable replies.*

---

### Official Review · Reviewer_czCd · 2025-10-23

**Soundness:** 1
**Presentation:** 3
**Contribution:** 2
**Rating:** 2
**Confidence:** 5

**Summary:**

The paper proposes a training-free plug-in that (1) reweights attention to emphasize a subset of "important" visual tokens while suppressing attention to system-prompt tokens, and (2) augments the standard Q–K attention with K–K similarity to make cross-modal attention more “semantically covariant.” Reported gains are shown on both yes/no VQA tasks (POPE) and generative benchmarks (CHAIR, AMBER).

**Strengths:**

1. The paper’s strength lies in its simplicity and efficiency. It introduces a lightweight mechanism that can enhance multimodal grounding with minimal cost.
2. The paper is clearly structured and provides sufficient implementation details to reproduce the method.

**Weaknesses:**

1. Narrow and observational motivation.
The central claim that “high-attention tokens should be amplified” is derived solely from Fig. 2(b) on POPE, a yes/no VQA dataset. This provides only correlational evidence on one task and does not establish a causal link to hallucination reduction. Although CHAIR and AMBER are later included, the motivation and design decisions are still primarily driven by POPE-style analysis.

2. Unproven assumption in SVE.
The SVE module ranks visual tokens by attention scores and assumes high attention equates to global information whose amplification reduces hallucination. This assumption is intuitive but unvalidated; the paper does not verify whether high-attention tokens correspond to true visual grounding or correct semantics.

3. Insufficient justification for system-prompt suppression.
The method penalizes attention to system-prompt tokens, yet the authors do not show that baseline models actually over-attend to these tokens or analyze the potential side effects of such suppression on instruction-following or safety.

4. Unclear necessity and risk in SCA design.
Using K–K (or V–V) similarity in cross-attention is argued to yield semantically aligned attention, but this modification may reduce diversity and introduce redundancy. The underlying problem in the base model is not well defined; if Q is indeed globally biased, a more direct and principled alternative would be to repair or adapt Q rather than replace Q–K with K–K.

5. Incomplete and potentially misleading AMBER evaluation.
The paper evaluates only on the generative subset of AMBER, ignoring the discriminative portion, and omits the “cover” metric. Without “cover”, models that output shorter or more conservative responses may appear artificially accurate, obscuring the true performance of the proposed method.

6. Limited robustness analysis of attention-layer choice.
The method ranks tokens using the penultimate-layer attention map, but it remains unclear whether the results hold for earlier or later layers. Testing layer robustness or multi-layer ensembling is necessary to confirm the stability of this design choice.

**Questions:**

1. How sensitive are the results to the hyperparameters controlling token reweighting strength and system-prompt suppression?
2. Have the authors tested the method’s robustness under adversarial or noisy visual inputs?

---

### Official Review · Reviewer_v3UY · 2025-10-31

**Soundness:** 3
**Presentation:** 3
**Contribution:** 2
**Rating:** 4
**Confidence:** 4

**Summary:**

The paper proposes VisionFocus, a training-free, plug-and-play method aimed at reducing hallucinations in MLLMs. The authors observe that in attention from the visual encoder tends to concentrate on a limited subset of tokens, and that indiscriminate enhancement of all visual tokens may worsen hallucination. The paper proposes two modules
* selective Visual Enhancement (SVE) – dynamically weight visual tokens by their attention importance, amplifying only high-attention tokens.
* semantic Covariant Attention (SCA) – in place of standard query-key attention, use key vectors as queries in local self-attention to promote semantic alignment and reduce invariance in semantics.

**Strengths:**

The authors recognise and analyse an important and practical problem: hallucinations in MLLMs.

The proposed solution is deployment-friendly (no retraining needed)

**Weaknesses:**

* Evaluation diversity — Although six benchmarks are used, most metrics center on CHAIR and POPE; inclusion of human evaluation or instruction-following datasets (e.g.,  MMHalBench/ hallucination bench / HPOPE, etc) could strengthen generality claims. . Without this broader, more fine‐grained benchmark suite the claim of “hallucination reduction” is less convincing, since the method may work for one type of hallucination but not generalise to others.

* Novelty is moderate — VisionFocus extends existing token-selection works (PAINT, Clearsight, VASparse) with refined weighting and semantic alignment, but the conceptual jump is evolutionary, not revolutionary. The correlation between attention tokens and hallucination is more of common sense, adjusting attention weights is not elegant eought for a ICLR paper.

* Limited theoretical depth — while the SCA mechanism is intuitively motivated, the paper lacks a formal analysis of why using keys as queries improves semantic localization.

**Questions:**

Please see weakness

---

### Official Review · Reviewer_YsD3 · 2025-11-01

**Soundness:** 2
**Presentation:** 3
**Contribution:** 2
**Rating:** 4
**Confidence:** 4

**Summary:**

The paper addresses the issue of hallucinations in Multimodal Large Language Models (MLLMs). The authors critique existing methods like contrastive decoding (for latency) and uniform visual attention enhancement (for sometimes worsening hallucinations). The core contribution is based on the insight that "Not all visual tokens are beneficial for hallucination mitigation". The authors observe that MLLM vision encoders naturally focus on a subset of high-attention visual tokens, which they find are crucial for grounding, while low-attention tokens can be distracting. Based on this, they propose "VisionFocus," a training-free and plug-and-play intervention that selectively enhances these "informative" visual tokens during decoding. The paper claims this method achieves state-of-the-art performance on six hallucination benchmarks while maintaining efficiency.

**Strengths:**

1.The paper correctly identifies a weakness in prior work, noting that indiscriminate enhancement of all visual tokens can be suboptimal or even harmful.
2.The proposed "VisionFocus" method is training-free and plug-and-play, which is a significant strength for practical application.
3.The paper highlights that its method maintains competitive decoding speed, avoiding the high latency associated with methods like contrastive decoding.

**Weaknesses:**

The paper operates on the flawed assumption that enhancing visual attention is the only variable that matters. It fails to discuss, analyze, or even acknowledge which tokens are being suppressed to pay for this "selective enhancement".
Also, the authors broadly categorize non-visual information as "language priors"  that should be suppressed. This ignores a growing body of research showing that specific tokens (e.g., in system prompts) are critical anchors for computational stability and function as part of the model's internal "state machine" for reasoning. If "VisionFocus" indiscriminately suppresses these tokens as part of its strategy, it is almost certainly destroying the model's higher-order reasoning capabilities.

**Questions:**

1.How exactly is the "selective enhancement"  implemented? Is this a pre-softmax score modification? If you boost the scores for "informative visual tokens," what (if any) is the corresponding modification to non-visual tokens (system prompts, user instructions, etc.)? Are they left alone, or are their scores suppressed?
2.The paper critiques suppressing "language priors". Did the authors perform any analysis on the different functional roles of non-visual tokens? Does your method treat a critical system-level token (like a BOS token or prompt anchor) the same as a token from the user's query?
3.Can the authors please provide evaluation results on comprehensive benchmarks that test multi-step reasoning, such as MM-Vet, SQA, and the MME benchmark (specifically, both its Perception and Reasoning subsets)? Without these, the claim of "effectiveness" is incomplete.
4.The method relies on identifying "informative visual tokens" , which are defined as those receiving "high attention". This seems circular. What is the precise, quantitative threshold for "high attention"? How does this method behave on simple images (e.g., "a red apple on a white background") where attention might be diffuse or all visual tokens might be "high-attention"? Does "VisionFocus" simply revert to the uniform enhancement it criticizes in such cases?

---

### Note · Authors · 2025-11-12

I have read and agree with the venue's withdrawal policy on behalf of myself and my co-authors.